# High Virulence and Multidrug Resistance of *Escherichia coli* Isolated in Periodontal Disease

**DOI:** 10.3390/microorganisms11010045

**Published:** 2022-12-23

**Authors:** Tania Hernández-Jaimes, Eric Monroy-Pérez, Javier Garzón, Rosario Morales-Espinosa, Armando Navarro-Ocaña, Luis Rey García-Cortés, Nancy Nolasco-Alonso, Fátima Korina Gaytán-Núñez, Moisés Moreno-Noguez, Felipe Vaca-Paniagua, Ernesto Arturo Rojas-Jiménez, Gloria Luz Paniagua-Contreras

**Affiliations:** 1Facultad de Estudios Superiores Iztacala, Universidad Nacional Autonoma de México, Los Reyes Iztacala, Tlalnepantla 54090, Mexico; 2Departamento de Microbiologia y Parasitologia, Facultad de Medicina, Universidad Nacional Autonoma de México, México City 04510, Mexico; 3Departamento de Salud Publica, Facultad de Medicina, Universidad Nacional Autonoma de México, México City 04510, Mexico; 4Instituto Mexicano del Seguro Social, Naucalpan de Juárez 53370, Estado de México, Mexico; 5Instituto Mexicano del Seguro Social, Tlalnepantla 54030, Estado de México, Mexico; 6Instituto Mexicano del Seguro Social, Los Reyes Iztacala, Tlalnepantla 54090, Estado de México, Mexico; 7Instituto Mexicano del Seguro Social, Zumpango 55600, Estado de México, Mexico; 8Unidad de Biomedicina, Facultad de Estudios Superiores Iztacala, Universidad Nacional Autonoma de México, Los Reyes iztacala, Tlalnepantla 54090, Mexico; 9Laboratorio Nacional en Salud, Diagnostico Molecular y Efecto Ambiental en Enfermedades Cronico-Degenerativas, Facultad de Estudios Superiores Iztacala, Universidad Nacional Autonoma de México, Los Reyes iztacala, Tlalnepantla 54090, Mexico; 10Subdireccion de Investigacion Basica, Instituto Nacional de Cancerologia, Mexico City 14160, Mexico

**Keywords:** periodontal disease, virulence markers, multidrug-resistance, phylogroups, serotypes

## Abstract

Periodontal disease is caused by different gram-negative anaerobic bacteria; however, *Escherichia coli* has also been isolated from periodontitis and its role in periodontitis is less known. This study aimed to determine the variability in virulence genotype, antibiotic resistance phenotype, biofilm formation, phylogroups, and serotypes in different emerging periodontal strains of *Escherichia coli*, isolated from patients with periodontal disease and healthy controls. *E. coli*, virulence genes, and phylogroups, were identified by PCR, antibiotic susceptibility by the Kirby-Bauer method, biofilm formation was quantified using polystyrene microtiter plates, and serotypes were determined by serotyping. Although *E. coli* was not detected in the controls (*n* = 70), it was isolated in 14.7% (100/678) of the patients. Most of the strains (*n* = 81/100) were multidrug-resistance. The most frequent adhesion genes among the strains were *fimH* and *iha*, toxin genes were *usp* and *hlyA*, iron-acquisition genes were *fyuA* and *irp2,* and protectin genes were *ompT,* and *KpsMT*. Phylogroup B2 and serotype O25:H4 were the most predominant among the strains. These findings suggest that *E. coli* may be involved in periodontal disease due to its high virulence, multidrug-resistance, and a wide distribution of phylogroups and serotypes.

## 1. Introduction

Periodontal disease is a chronic inflammatory disease caused by the accumulation of microorganism biofilms around the teeth, which promotes the formation of periodontal pockets, which over time cause the progressive destruction of the supporting structures of the teeth, potentially leading to tooth loss [1]. Among the bacteria that cause periodontal disease are gram-positive, and gram-negative anaerobic bacteria such as *Porphyromonas gingivalis*, *Prevotella intermedia*, *Aggregatibacter actinomycetemcomitans*, *Treponema denticola*, and *Tannerella forsythia* [2]; other gram-positive facultative anaerobic bacteria, such as *Staphylococcus aureus* [3] and *Escherichia coli* have also been identified as colonizer related with periodontitis [4]. The pathogenicity of *E. coli* during periodontal disease is unknown, however it could contribute to the disease due to its multiple virulence-associated factors, including adhesin-coding genes (*fim*, *afaI*, *sfa*, *iha*, *tsh*, *papC*, *papGI*, *II*, and *III*) that favor the colonization and formation of biofilms, toxins *(cnf1*, *hlyA*, *set-1*, *astA*, *vat*, *usp*, and *cva/cvi*), iron acquisition systems (*iroN*, *irp2*, and *iuc*), and protectin genes (*kpsMT*, *ompT*, and *iss*), and the horizontal transfer of many of these genes between bacteria occurs via pathogenicity-associated islands (PAIs) [5,6]. In this regard, biofilm production by *E. coli* may play an important role in the persistence of bacterial infections by conferring antimicrobial tolerance and increased resistance against the host defense [7]. The biofilms associated with periodontal disease have been described as a reservoir of medically important pathogen species commonly associated with relevant systemic infections, including enterobacteria, *C. difficile*, *P. aeruginosa*, *A. baumannii*, *N. gonorrhoeae*, *C. albicans*, *S. aureus*, *H. pylori*, *G. vaginalis*, *S. marcescens*, and *S. pneumoniae*, among others [8].

Currently, the distribution of the different phylogroups and serotypes in *E. coli* associated with periodontal disease have not been studied, however, phylogenetic analyzes of *E. coli* strains causing other extraintestinal infections have classified this species into eight major phylogroups, seven belonging to *E. coli* sensu stricto (A, B1, B2, C, D, E, and F) and one to clade I (a cryptic *E. coli* phylogroup) [9]. Serotyping is one of the most frequently used traditional methods for the classification of *E. coli*, which relies on antisera to identify somatic O antigens and flagellar H antigens [10]. Approximately 187 O antigens and 53 H antigens have been reported in *E. coli* so far [11].

The treatment of infections caused by *E. coli* is currently a public health problem, due to the emergence of multi-resistant *E. coli* strains to multiple families of antibiotics [12,13]. 

Virulence factors in anaerobic bacteria that cause periodontal disease have been studied. Relevant examples include *Porphyromonas gingivalis*, which produces several virulence factors in the outer cell membrane, including gingipains and hemagglutinins, that enable the bacteria to adhere to periodontal tissue and degrade host proteins to obtain the nutrients needed for dental plaque formation [14]. *Prevotella intermedia* produces two proteins homologous to *Porphyromonas gingivalis* HmuY, but with different heme coordination mode [15]. *Aggregatibacter actinomycetemcomitans* produces a leukotoxin, which induces apoptosis in human leukocytes [16]. *Treponema denticola* produces the protease dentilisin, which is involved in nutrient uptake, bacterial coaggregation, complement activation, the evasion of the host immune system, the inhibition of the hemostasis system, and cell invasion [17]; *Tannerella forsythia* is a bacterial pathogen implicated in periodontal disease that has numerous genes associated with virulence [18].

The virulence factors of *E. coli* such as adhesins, iron uptake systems, toxins and protectins have been studied in uropathogenic strains [19], but the molecular characteristics of *E. coli* and its role in periodontitis remain elusive. Therefore, the aim of this study was to determine the variability in the virulence genotype, antibiotic resistance phenotype, biofilm formation, phylogroups, and serotypes in the strains of *E. coli* isolated from patients with periodontal disease.

## 2. Materials and Methods

### 2.1. Patient Selection and Isolation of E. coli

A total of 678 patients (365 women and 313 men) with periodontal disease were selected from the dental service of Hospital No. 60 of the Mexican Institute of Social Security (*Instituto Mexicano del Seguro Social*—IMSS), State of Mexico. Controls, a group of healthy volunteers (n = 70) who did not exhibit clinical signs of periodontitis, were also included in the study. All subjects gave their informed consent for inclusion before they participated in the study. The study was conducted in accordance with the Declaration of Helsinki, and the protocol was approved by the Ethics Committee of the institution (Project identification code: R-2022-1406-034). Patients who were smokers, underwent periodontal treatment, or were treated with antibiotics in the last 6 months were excluded from this study. The sample group included patients with gingivitis (only where irritation, redness, and inflammation in the gums were displayed), in addition to patients with periodontitis who presented periodontal pockets of more than 4 mm. Patients with moderate periodontitis had dental insertion loss of less than 4 mm, and patients with chronic periodontitis greater than 5 mm. Sterile swabs were used to collect samples from patients. Samples were taken from the surface of the marginal gingiva in patients with gingivitis and by pressing the periodontal pocket in patients with periodontitis. The samples were cultured immediately in 2 mL of Brain Heart Infusion (BHI) broth (Bioxon, México City, México), and incubated at 37 °C for 24 h. Subsequently, the cultures were reseeded by the cross-streak method in the solid culture medium, eosin methylene blue (EMB; MCD LAB, Tlalnepantla, Edo. de México), S110 (MCD LAB, Tlalnepantla, Edo. de México), and Sabouraud (MCD LAB, Tlalnepantla, Edo. de México), and incubated under aerobics conditions at 37 °C for 24 h. The *E. coli* strains were identified by standard IMViC biochemical tests (Indole, Methyl Red, Voges-Proskauer and Citrate). *S. aureus* strains were identified by bacteriological and biochemical tests that included mannitol, coagulase, and polymerase chain reaction (PCR) by amplification of the 23S rRNA, *nuc*, *spa* x region, and *coa* genes [20]. *C. albicans* was identified by PCR amplification of internal transcribed spacers (ITS) 1 and 2 of the rRNA gene [21], *S. epidermidis* was identified by PCR as previously described [22], and, *K. pneumoniae* was identified by PCR based on the 16S-23S rDNA internal transcribed spacer [23]. 

### 2.2. DNA Extraction and Identification of E. coli by Polymerase Chain Reaction (PCR)

Bacterial DNA was extracted as previously described [24]. From a pure culture of each *E. coli* strain identified by standard IMViC biochemical tests, a colony was taken, reseeded in 2 mL of Brain Heart Infusion (BHI) broth (Bioxon, México city, México) and incubated at 37 °C for 12 h. The cell pellet was obtained by centrifugation and suspended in 200 μL of sterile water. The resuspended pellet was incubated at 100 °C for 10 min and centrifuged at 10,000× *g* for 5 min. The pellet was then discarded, and the DNA present in the supernatant was stored at ~20 °C until testing. *E. coli* strains were identified by PCR amplification of the 16S rRNA gene [25]. Stock dilutions of genomic DNA from each *E. coli* strain were prepared with nuclease-free water to obtain concentrations of 100 ng/μL. The concentration and purity of the DNA was measured in a NanoDrop 2000 spectrophotometer (Thermo Fisher Scientific, Waltham, MA, USA). Each PCR assay was developed using 20 μL of the reaction mix, which included 12 μL Taq DNA Polymerase 2X Master Mix RED (Ampliqon), 1 μL forward primer (F- AGAGTTTGATCMTGGCTCAG), 1 μL reverse primer (R-CCGTCAATTCATTTGAGTTT; Integrated DNA Technologies), 3 μL nuclease-free water, and 3 μL DNA template (300 ng). The concentration of the primers used was 10 pmol. The amplification conditions consisted of initial denaturation for 5 min at 95 °C, 30 cycles of denaturation at 95 °C for 30 s, alignment at 55 °C for 1 min, and extension at 72 °C for 1 min, and a final extension at 72 °C for 5 min. *E*. *coli* ATCC 11775 was used as the positive control for each PCR assay [25]. All the primers, and genes evaluated by PCR in this study are presented in Appendix A.

### 2.3. Antibiotic Susceptibility Test

The Kirby-Bauer disk diffusion method was used to test the following 12 antibiotics: ampicillin (AM; 10 μg), carbenicillin (CB; 100 μg), cephalothin (CF; 30 μg), cefotaxime (CFX; 30 μg), ciprofloxacin (CPF; 5 μg), chloramphenicol (CL; 30 μg), nitrofurantoin (NF; 300 μg), amikacin (AK; 30 μg), gentamicin (GE; 10 μg), netilmicin (NET; 30 μg), norfloxacin (NOF; 10 μg), and sulfamethoxazole trimethoprim (SXT; 25 μg) (Investigación Diagnóstica, México City, México). These antibiotics are frequently used by the public health sector in Mexico. The results were interpreted according to the criteria established by the Clinical and Laboratory Standards Institute [26]. The American Type Culture Collection (ATCC) strain *E. coli* 25922 was used as the control. Strains resistant to more than three antibiotics belonging to different categories were classified as multidrug-resistant (MDR) [27].

### 2.4. Virulence Factor Detection

Previously described PCR conditions and primers were used to amplify the genes encoding adhesins (*papA*, *papGI*, *papGII*, *papGIII*, *fimH*, *afa*, *sfaS*, *iha*, *focG*, and *bmaE*), toxins (*cnf1*, *hlyA*, *tsh*, *usp*, *set 1* and *astA*), iron acquisition systems (*iuc*, *iroN*, *irp2*, *feoB*, *fyuA*, and *ireA*), protectins (*kpsMT*, *ompT*, *iss*, and *traT*), and pathogenicity islands (*malX*) [28,29]. Each uniplex PCR assay was developed using 20 μL of the reaction mix that included 12 μL Taq DNA Polymerase 2X Master Mix RED (Ampliqon), 1 μL forward primer, 1 μL reverse primer (Integrated DNA Technologies), 3 μL nuclease-free water, and 3 μL DNA template (300 ng). The concentration of the primers used was 10 pmol.

### 2.5. Quantification of Biofilm Formation

Biofilms produced by *E. coli* were quantified in triplicate on 96-well polystyrene microtiter plates according to the method as previously described [30]. Adhered bacterial cells were stained with 0.1% crystal violet for 15 min, followed by washing three times with PBS. Bound bacteria were quantified by adding dimethyl sulfoxide at an optical density of 595 nm in an ELISA plate reader (Multiscan, Labsystems, Helsinki, Finland). *E. coli* strain ATCC 25922 was used as the positive control. Biofilm formation capability was calculated as follows: BF = AB/CW, with BF = biofilm formation; AB = stained attached bacteria; CW = stained control wells [30]. 

Interpretive criteria were as follows: ≥6.00 = strongly positive; 4.00–5.99 = moderately positive; 2.00–3.99 weakly positive; <2.00 = non-biofilm. 

### 2.6. Identification of Phylogroups

The multiplex PCR method and conditions used to identify phylogroups A, B1, B2, C, D, E, F, and cryptic clade I were previously described [9]. This method allows for the identification of different phylogroups by amplifying *chuA* and *yjaA*, the DNA fragments *TspE4.C2*, and *arpA.* The final multiplex PCR reaction volume was 20 μL, which included 9 μL of Taq DNA Polymerase 2× Master Mix RED (Ampliqon), 1 μL of first forward and reverse primers, and 3 μL of DNA template (300 ng). The concentration of the primers used was 10 pmol. To determine groups E and C, singleplex PCR was performed separately. The PCR conditions were as follows: 4 min at 94 °C, 30 cycles of 5 s at 94 °C, and 20 s at 57 °C (group E) or 59 °C (group C), and a final extension of 5 min at 72 °C [9]. 

### 2.7. Serotyping

The antigens were characterized according to the description by Orskov et al. [31], using 186 rabbit sera against the somatic antigen (O) and 53 rabbit sera against the flagellar antigen (H) (SERUNAM, Mexico). Agglutination reactions were carried out in triplicate on 96-well microtiter plates. The serotype of the strains was determined by the O:H antigenic formula obtained after evaluating the agglutination titers of the sera used.

### 2.8. Statistical Analysis

We used the Chi-square test with the SPSS statistical program (version 20.0; SPSS Inc., Chicago, IL, USA) (*p* <0.05) [32] to establish the differences between the percentages of resistance to antibiotics, virulence genes, biofilm formation, phylogroups, serotypes, and different diagnoses of periodontal diseases. 

### 2.9. Unsupervised Hierarchical Clustering

To fully visualize the global characteristics of the strains, we used unsupervised hierarchical grouping with Gower’s similarity algorithm for categorical variables to group them according to their molecular characteristics [33]. A categorical data matrix, including virulence genes, phylogroups, serotypes, biofilm production, and type of infection (gingivitis, moderate periodontitis, and advanced periodontitis), was created in R (v3.6.1) using the cluster package (2.1.0). The distance of each strain was calculated based on the general similarity coefficient, which estimates the maximum possible absolute discrepancy between each combined pair of strains. After calculating the distances, mutually exclusive groups were grouped using Ward’s method in R [34]. The strains were visualized in a genotype-phenotype distribution diagram with a dendrogram constructed using hclust (v3.6.2, R core). The R script and matrices used are presented in Appendix A.

## 3. Results

### 3.1. Patients Studied

Of the 678 patients with periodontal disease, 14.7% (n = 100) of the samples were identified by PCR. Most *E. coli* strains were isolated from patients with moderate periodontitis (n = 59/100), followed by chronic periodontitis (n = 29/100), and gingivitis (n = 12/100) (Table 1). In addition, we identified other bacteria associated with *E. coli* (n=100) in periodontal infections, including *E. coli*/*Staphylococcus epidermidis* in 31% of cases, *E. coli/Staphylococcus aureus* in 23%, *E. coli/Candida albicans* and *E. coli/Staphylococcus epidermidis/Candida albicans* in 3%, and *E. coli/Staphylococcus aureus/Candida albicans*, and *E. coli/Klebsiella pneumoniae* in 1% of cases. Interestingly, in the group of healthy individuals (n = 70), *E. coli* was not detected in any of the subjects; however, *Staphylococcus epidermidis* and *Staphylococcus aureus* were identified in 41.4% of the cases, and *Candida albicans* in 1.4 %.

### 3.2. Multiresistance to Antibiotics in E. coli Strains

The highest antibiotic resistance rates of *E. coli* strains were for the beta-lactams ampicillin, carbenicillin, cephalothin, and cefotaxime, as well as for nitrofurantoin (Table 1). Antibiotic resistance percentages were similar in *E. coli* strains isolated from patients with gingivitis, moderate periodontitis, and chronic periodontitis, except for the percentage of resistance to chloramphenicol, which was higher in the strains of patients with gingivitis (*p* < 0.05, Table 1). 

Most of the strains (n = 81) were multiresistant to antibiotics, with 33% being multiresistant to four antibiotics and 19% to five antimicrobials.

### 3.3. Detection of Virulence Genes Related to Phylogroups

The components of the virulence genes were found to be widely distributed among *E. coli* strains isolated from patients with periodontal disease (Table 2). Half (n = 50) of the *E. coli* strains were classified into one of the seven phylogroups studied (A1, B1, B2, C, D, E, and F). The most prevalent phylogroups were B2 (n = 26), E (n = 11), and B1 (n = 7), which were more frequent. There were significant differences between the frequency of the adhesion genes *fimH*, *iha*, *papA*, *sfaS*, *papGIII*, and *focG* in the *E. coli* strains assigned and not assigned to the seven phylogroups according to the three diagnoses (*p* < 0.05, Table 2), and were identified more frequently in strains belonging to phylogroup B2 from patients with moderate periodontitis than in strains from patients with chronic periodontitis or gingivitis. Similarly, the toxin genes *usp*, *hlyA*, *cnf1*, and *set-1*, the iron acquisition genes *fyuA* and *irp2*, and the protective genes *ompT* and *kpsMT* were detected more frequently in the strains belonging to phylogroup B2 from patients with moderate periodontitis (*p* < 0.05, Table 2). The *malX* pathogenicity island marker was more prevalent in strains with phylogroup B2 isolated from patients with moderate periodontitis than in strains isolated from patients with other diagnoses (*p* < 0.05).

The 100 *E. coli* strains analyzed were distributed within 80 different combinations of virulence genes related to phylogroups and periodontal disease (Table 3). Thirteen strains were distributed among the gene combinations from number 1 to 6, in which combination No. 1, characterized by *feoB/iroN* iron acquisition genes, was found in three strains related to moderate periodontitis, one belonging to phylogroup B2, and two with unidentified phylogroup. The combination No. 4 composed of genes that code for adhesins, iron uptake systems, toxins, protectins, and pathogenicity islands was found in two strains, one belonging to phylogroup B2 (moderate periodontitis) and the other to phylogroup D (chronic periodontitis). The virulence gene combinations of No. 7 to 80 were composed by 87 strains, each one having a different combination of the same genes, (data not shown). 

### 3.4. Identification of Virulence Genes Related to Biofilm Formation

The majority of the strains (n = 89) produced biofilms (Table 4). The prevalence of virulence genes was similar between biofilm producer strains (weak, moderate, and strong) and non-biofilm producer strains (n = 11; *p* < 0.05), except for *papGIII*, which was more prevalent in non-biofilm producer strains, and *hlyA*, which was more prevalent in strong biofilm-producing strains (*p* < 0.05). *papA*, *sfaS*, *papGII*, *focG*, *hlyA*, and *cnf1* were identified in strains that produced biofilms but not in strains that did not produce biofilms.

### 3.5. Identification of Serotypes in Strains Related to Phylogroups

A total of 41 different *E. coli* serotypes were identified, the majority of which (n = 59) were isolated from patients with moderate periodontitis (Table 5). The most prevalent serotype was O25:H4 (n = 14), predominantly identified in strains isolated from patients with chronic and moderate periodontitis, whereas O12:H- (n = 12) was more common in strains of patients with moderate periodontitis and gingivitis. There were no significant differences between the frequency of the serotypes associated with the different phylogroups of strains isolated from patients of the three diagnoses (gingivitis, moderate periodontitis, and chronic periodontitis; *p* < 0.05), except for the O2: H39 serotype, which was more prevalent in the B2 phylogroup of strains with moderate periodontitis and in serotype O15: H18, which was more prevalent in the phylogroup E of strains with moderate periodontitis (*p* < 0.05).

### 3.6. Genotypic and Phenotypic Diversity

To characterize the genotypic and phenotypic diversity globally, we performed unsupervised hierarchical clustering analysis. Two large groups were identified based on the similarities between the *E. coli* strains according to the virulence genotype profile and its relationship with biofilm production, phylogroups, serotypes, and diagnoses (Figure 1). Group 1, which was divided into several subgroups, consisted of 62 strains (I167 to U207; Figure 1), while group 2 consisted of 38 strains (I261 to U115). Most of the strains (n = 80) presented a different virulence genotype composed of different arrays of genes, and 20% of the strains (n = 20) had identical genotype (100% similarity) in both groups. For example, strains U1 (phylogroup B2; serotype O6:H1; strong biofilm) and U10 (phylogroup D; serotype O2:H39; moderate), which were isolated from patients with moderate and chronic periodontitis, respectively, had the same genotype pattern, consisting of 19 virulence genes. Other pairs of strains (U21 and U51; U88 and U115) with identical genotype (100% similarity) were also identified in the subgroups of group 2 (Figure 1).

## 4. Discussion

Although a variety of gram-negative anaerobic bacteria cause periodontal disease [35], other pathogenic non-oral bacteria, such as *E. coli*, have been isolated from patients with periodontal disease [36]. *E. coli* was isolated in 14.7% (n = 100/678) of patients with periodontal disease in this study, most frequently in those with moderate periodontitis (n = 59; Table 1) rather than in those with chronic periodontitis or gingivitis. This proportion is lower than that reported in other studies of patients with chronic periodontitis or aggressive periodontitis [36,37]. Most of the *E. coli* strains isolated from patients with periodontal disease were multiresistant to groups of three–six antibiotics (n = 81, Table 1). The highest percentages of resistance were found for the beta-lactams ampicillin, carbenicillin, cephalothin, and cefotaxime, as well as for nitrofurantoin, which were similar between the *E. coli* strains isolated from patients with gingivitis, moderate periodontitis, and chronic periodontitis (*p* < 0.05, Table 1). High resistance to beta-lactam antibiotics, primarily AM, has also been reported in strains of *E. coli* isolated from patients with periodontitis and gingivitis [38]. In a recent study, we discovered comparable resistance rates to AM, CB, CF, NET, and GE in cervicovaginal strains of *E. coli* [39]. The high resistance rate of strains to beta-lactam antibiotics found in this study may be due to the substantial increase in *E. coli*-producing extended-spectrum beta-lactamase (ESBL), both within the community and in hospitals [40].

The *E. coli* strains analyzed belonged primarily to phylogroups B2, E, B1, and A (Table 2). These phylogroups were recently identified in *E. coli* strains that cause urinary tract infections [41] and in *Escherichia coli* cervicovaginal strains [42]. These findings demonstrate, for the first time, the broad phylogenetic versatility of *E. coli* isolated from patients with periodontal infections. To date, no study has been conducted on the distribution of virulence markers among the various phylogroups of *E. coli* strains isolated from patients with periodontal disease. Because adhesion promotes colonization and invasion of tissues, it is considered the most important pathogenic factor for *E. coli* [43]. The most frequently identified adhesion genes were *fimH*, *iha*, *papA*, *sfaS*, *papGIII*, and *focG* (Table 2), which were more prevalent in strains belonging to phylogroup B2 from patients with moderate periodontitis than in strains belonging to phylogroup B2 from patients with chronic periodontitis or gingivitis (*p* < 0.05, Table 2).

The prevalence of these genes in *E. coli* may favor the persistence of periodontal disease, since the expression of *fim* and *sfaS* has been associated with the formation of biofilms in uropathogenic *E. coli* (UPEC) strains [44], and has been frequently identified in *E. coli* strains causing cystitis [45]. *papA* genes and the allelic variants *papGI* and *papGII* encoding P fimbria (Table 2) cause other types of infections, as these genes are frequently identified in *E. coli* uropathogenic strains that cause cystitis, prostatitis, and pyelonephritis [46].

Toxins are clinically relevant molecules because they cause tissue damage in the host, facilitate nutrient release and bacterial spread, and inactivate effector cells of the immune system [47]. The toxin genes *usp*, *hlyA*, *cnf-1*, and *set-1* were more frequent in strains of phylogroup B2 in patients with moderate periodontitis (*p* < 0.05, Table 2). The prevalence of these virulence markers in the strains, in addition to the presence of the *malX* marker of the PAIs (Table 2) may increase the risk of acquiring other types of infections. This is evident by the fact that we identified these genes in strains of *E. coli* responsible for cervicovaginal infections [39], whereas other authors have identified them in UPEC strains causing urinary tract infections [48]. 

The iron acquisition genes *fyuA* and *irp2*, and the protectins *ompT* and *KpsMT* were prevalent among the strains of phylogroup B2 in patients with moderate periodontitis (*p* < 0.05, Table 2), while the frequency of the *feoB* and *traT* genes was similar between the phylogroups of the strains isolated from patients with the three diagnoses (*p* < 0.05). The presence of these virulence markers in *E. coli* may increase periodontal disease pathogenicity, as iron is essential for the survival and persistence of *E. coli* in infectious processes [49], while the capsular antigen K encoded by *kpsMT* protects the bacteria against phagocytosis, and *traT* and *iss* increase resistance to serum [50]. Iron acquisition and protectin genes have been identified in strains of extraintestinal *E. coli* that cause urinary tract infections [51] and septicemia [50].

Numerous patterns of virulence markers were identified in *E. coli* strains related to the phylogroups and periodontal disease of the patients (n = 80; Table 3). Pattern 4 contained a greater number of genes (n = 19) and was detected in a strain belonging to phylogroup B2 (moderate periodontitis) and the other to phylogroup D (chronic periodontitis; Table 3); Pattern 5 contained 15 genes and was found in a strain belonging to phylogroup B2 from patients with gingivitis and moderate periodontitis. The coexistence of different combinations of virulence markers identified in *E. coli* strains may favor the acuteness of periodontal infections.

Most of the strains (n = 89) formed biofilms, predominantly in patients diagnosed with moderate periodontitis (n = 33; Table 4). The frequency of virulence genes was comparable between biofilm-producing strains (weak, moderate, and strong) and non-biofilm producing strains (n = 11; *p* < 0.05), but the genes *papA*, *sfaS*, *papGII*, *focG*, *hlyA*, and *cnf1* were identified exclusively in the biofilm-producing strains. These findings demonstrate that *E. coli* strains are capable of contributing to the pathogenicity of chronic periodontal infections, as the biofilm protects the bacteria from the host’s immune response and antibiotic action [52]. Notably, the percentages of biofilm production identified in the strains were comparable to those previously reported in highly pathogenic UPEC strains [44]. Additionally, it has been suggested that *E. coli* lipopolysaccharides (LPS) may have a role in the inflammatory processes associated with periodontal disease [53,54].

In this study, 41 distinct serotypes of *E. coli* strains isolated from periodontal disease were identified using serological methods (Table 5). The serotype distribution was similar across strain phylogroups (*p* < 0.05), except for the O2: H39 serotype, which was more prevalent in the B2 phylogroup of strains from patients with moderate periodontitis, and in serotype O15: H18, which was more prevalent in the E phylogroup of strains from patients with moderate periodontitis (*p* < 0.05). The most frequently encountered serotype was O25: H4, which was more prevalent in strains belonging to phylogroup B2, and strains that did not belong to any of the phylogroups studied in patients with chronic and moderate periodontitis (Table 5). These results show that many serotypes that cause other types of infections may participate in the pathogenesis of periodontal disease, as was the case for serotype O25:H4, which has been frequently associated with uropathogenic strains responsible for urinary tract infections [55]. 

*E. coli* strains isolated from the three clinical diagnoses were classified into two large groups, which were further divided into subgroups according to their virulence genotype, biofilm formation phenotype, phylogroups, and serotypes (Figure 1). The strains within group 1 (strains I167 to U207; see Figure 1) were predominantly serotype O24H:4 and belonged to phylogroup B1, while the strains in group 2 (strains I261 to U115) had a greater number of virulence markers and most frequently belonged to phylogroup B2 and serotype O25:H4 and O12:H-. It is worth noting that most of the strains presented a different virulence genotype, which leads us to assume that these strains isolated from patients with periodontal disease are different. However, strains with identical virulence genotype were also identified in unrelated patients belonging to some of the subgroups of groups 1 and 2, which suggests that these strains may be prevalent in the population with periodontitis (Figure 1). Interestingly, *E. coli* was not detected in any of the subjects in the group of healthy individuals, which suggests a pathogenic role of *E. coli* that needs to be studied further. Overall, these results demonstrate that complex and diverse expression patterns of virulence genes of *E. coli* may occur during the pathogenesis of periodontal disease, which could lead to more acute or chronic infections when combined with multiresistance to antibiotics and biofilm production. We could not identify the phylogroup in half of the *E. coli* strains, possibly due to variation in primer recognition sites or due to genetic recombination events, which may have prevented amplification [56]. It is also possible that these strains belong to very rare phylogroups that are not covered by this method [57]. The same occurred in some of the serotypes, which could probably be due to new serotypes. Therefore, further studies of whole genome sequencing of these strains could provide precise identification of their phylogroups, and serotypes, as well as their virulence and antibiotic resistance gene load.

## 5. Conclusions

In summary, the findings of this study are significant as they reveal the wide molecular variation of *E. coli* that may act as a pathogen in periodontal disease. We found that the strains harbored a substantial number of virulence markers and notable multiresistance to the antibiotics commonly used in clinical practice. The characterization of the strains showed that phylogroup B2 and serotype O25:H4 were the most prevalent. This is the first global molecular study conducted on periodontal strains of *E. coli* and can aid in the development of new strategies for the treatment of periodontal infections in the future. 

## Figures and Tables

**Figure 1 microorganisms-11-00045-f001:**
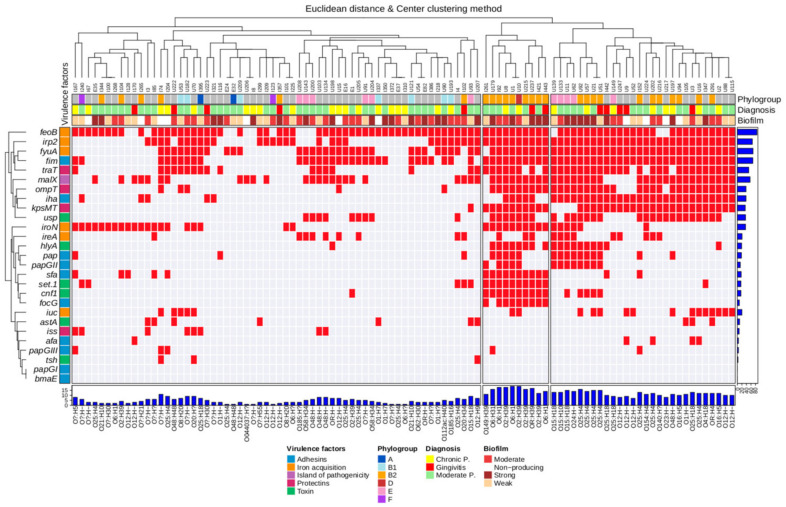
Hierarchical clustering of *E. coli* strains according to the virulence genotype profile and its relationship with biofilm production, phylogroups, serotypes, and diagnoses. Positivity and negativity for a given genotype are represented by a red and gray “rectangle” respectively. Genetic functions, biofilm production, phylogroups, serotypes, and diagnostics are color-coded as indicated in the key below. Cladograms of the strains and genes are shown along the top and “left” respectively. Upper axis: ID of the strains. Left axis: name of the virulence genes. Right blue bars: detection percentage of each virulence gene. Lower blue bars: number of virulence genes detected in each strain. Lower axis: serotypes identified in each strain.

**Table 1 microorganisms-11-00045-t001:** Resistance and multiresistance to antibiotics in *E. coli* (n = 100).

Antibiotic Group	Antibiotics	No. of Resistant Strains (%)			
		Gingivitis (n = 12) No. (%)	Moderate Periodontitis (n = 59) No. (%)	Chronic Periodontitis (n = 29) No. (%)	*p*-Value
Beta-lactams	Ampicillin (AM)	12 (100)	57 (97)	29 (100)	1
	Carbenicillin (CB)	11 (92)	55 (93)	28 (97)	0.84
	Cephalothin (CF)	11 (92)	50(85)	28 (97)	0.29
	Cefotaxime (CFX)	9 (75)	34 (58)	17 (59)	0.60
Quinolones	Norfloxacin (NOF)	5 (42)	27 (46)	15(52)	0.8
	Ciprofloxacin (CPF)	5 (42)	26 (44)	13 (45)	0.98
Phenicols	Chloramphenicol (CL)	7 (58)	10 (17)	8 (28)	**0.01**
Nitrofurans	Nitrofurantoin (NF)	10 (83)	50 (85)	24 (83)	1
Aminoglycosides	Amikacin (AK)	4 (33)	35 (59)	18 (62)	0.2
	Gentamicin (GE)	6 (50)	28 (47)	9 (31)	0.29
	Netilmycin (NET)	6 (50)	28 (47)	17 (59)	0.61
Sulfonamide/Trimethoprim	Sulfamethoxazole/trimethoprim	8 (67)	25 (42)	14 (48)	0.3
Multiresistance (Different Families of Antibiotics) (n = 81)		No. of Antibiotic Groups			No. of Strains
		3			18
		4			33
		5			19
		6			11

Notes: Significant *p*-values (<0.05) were presented in bold.

**Table 2 microorganisms-11-00045-t002:** Frequency of virulence genes according to phylogroups in *E. coli*.

Function	Phylogroups	Gingivitis (n = 12) No. of Strains									Moderate Periodontitis (n = 59) No. of Strains									Chronic Periodontitis (n = 29) No. of Strains									*p*-Value	Total (n = 100)
		A	B1	B2	C	D	E	F	CLADE I	Unidentified	A	B1	B2	C	D	E	F	CLADE I	Unidentified	A	B1	B2	C	D	E	F	CLADE I	Unidentified		
	Gene	0	0	5	0	0	0	0	0	7	2	5	15	0	0	9	2	0	26	1	2	6	0	1	2	0	0	17		
Adhesins	*fimH*	-	-	3	-	-	-	-	-	5	1	4	12	-	-	9	1	-	13	-	2	6	-	1	2	-	-	9	**0.005**	68
	*iha*	-	-	2	-	-	-	-	-	5	-	-	9	-	-	3	1	-	8	-	-	5	-	1	2	-	-	2	**0.005**	38
	*papA*	-	-	2	-	-	-	-	-	-	-	-	6	-	-	3	-	-	1	-	-	3	-	1	1	-	-	1	**0.0000887**	18
	*sfaS*	-	-	2	-	-	-	-	-	1	-	-	8	-	-	-	-	-	2	-	-	1	-	1	-	-	-	2	**0.002**	17
	*papGII*	-	-	1	-	-	-	-	-	-	-	-	5	-	-	3	-	-	-	-	-	2	-	1	1	-	-	-	**0.0000292**	13
	*focG*	-	-	2	-	-	-	-	-	-	-	-	5	-	-	-	-	-	-	-	-	1	-	1	-	-	-	-	**0.0001**	9
	*afa*	-	-	-	-	-	-	-	-	3	-	-	-	-	-	-	-	-	-	-	-	2	-	-	-	-	-	-	1	5
	*papGIII*	-	-	-	-	-	-	-	-	-	-	-	2	-	-	-	-	-	-	-	-	1	-	-	-	-	-	2	0.65	5
	*papGI*	-	-	-	-	-	-	-	-	-	-	-	-	-	-	-	-	-	-	-	-	-	-	-	-	-	-	-	0	0
	*bmaE*	-	-	-	-	-	-	-	-	-	-	-	-	-	-	-	-	-	-	-	-	-	-	-	-	-	-	-	0	0
Toxins	*usp*	-	-	2	-	-	-	-	-	1	-	-	10	-	-	6	-	-	7	-	-	5	-	1	1	-	-	3	**0.0000467**	36
	*hlyA*	-	-	3	-	-	-	-	-	1	-	-	6	-	-	3	-	-	-	-	-	2	-	1	1	-	-	2	**0.0003**	19
	*cnf1*	-	-	3	-	-	-	-	-	-	-	-	8	-	-	1	-	-	1	-	-	2	-	1	-	-	-	-	**0.0000025**	16
	*set-1*	-	-	3	-	-	-	-	-	-	-	-	6	-	-	1	1	-	2	-	-	1	-	1	-	-	-	1	**0.001**	16
	*astA*	-	-	-	-	-	-	-	-	2	-	-	1	-	-	1	-	-	4	-	1	-	-	-	-	-	-	1	0.73	10
	*tsh*	-	-	-	-	-	-	-	-	-	-	-	-	-	-	-	-	-	1	-	-	1	-	-	-	-	-	1	1	3
Iron Acquisition	*feoB*	-	-	5	-	-	-	-	-	6	1	5	14	-	-	6	1	-	23	1	2	6	-	1	1	-	-	14	0.05	86
	*fyuA*	-	-	3	-	-	-	-	-	6	2	5	11	-	-	9	-	-	14	-	1	6	-	1	2	-	-	9	**0.01**	69
	*irp2*	-	-	5	-	-	-	-	-	5	1	3	12	-	-	4	1	-	15	-	1	6	-	1	2	-	-	10	0.05	66
	*iroN*	-	-	2	-	-	-	-	-	1	1	1	10	-	-	3	1	-	8	-	-	2	-	1	1	-	-	5	0.15	36
	*ireA*	-	-	2	-	-	-	-	-	1	-	-	3	-	-	6	-	-	4	-	-	1	-	-	2	-	-	2	**0.002**	21
	*iuc*	-	-	1	-	-	-	-	-	3	-	2	4	-	-	-	-	-	4	-	-	3	-	1	-	-	-	2	0.13	20
Protectins	*traT*	-	-	3	-	-	-	-	-	6	1	3	10	-	-	5	-	-	8	-	-	5	-	1	2	-	-	6	0.08	50
	*ompT*	-	-	3	-	-	-	-	-	2	-	3	11	-	-	3	-	-	5	-	-	6	-	1	2	-	-	2	**0.00000345**	38
	*kpsMT*	-	-	3	-	-	-	-	-	4	-	-	12	-	-	1	-	-	6	-	-	5	-	1	1	-	-	2	**0.00000283**	35
	*iss*	-	-	-	-	-	-	-	-	-	1	1	-	-	-	-	1	-	3	-	-	-	-	-	-	-	-	2	0.05	8
PAIs	*malX*	-	-	4	-	-	-	-	-	2	-	1	13	-	-	7	-	-	13	-	-	6	-	1	1	-	-	8	**0.0000159**	56

Notes: Significant *p*-values (<0.05) were presented in bold.

**Table 3 microorganisms-11-00045-t003:** Distribution of virulence gene association patterns in strains of *E. coli*.

No. of Combination	Combination of Virulence Genes	Gingivitis (n = 12) No. of Strains									Moderate Periodontitis (n = 59) No. of Strains									Chronic Periodontitis (n = 29) No. of Strains									Total (n = 100)
		A	B1	B2	C	D	E	F	CLADE I	Unidentified	A	B1	B2	C	D	E	F	CLADE I	Unidentified	A	B1	B2	C	D	E	F	CLADE I	Unidentified	
		0	0	5	0	0	0	0	0	7	2	5	15	0	0	9	2	0	26	1	2	6	0	1	2	0	0	17	
1	*feoB/iroN*	-	-	-	-	-	-	-	-	-	-	-	1	-	-	-	-	-	2	-	-	-	-	-	-	-	-	-	3
2	*feoB/irp2/iroN*	-	-	-	-	-	-	-	-	-	-	-	-	-	-	-	-	-	1	-	-	-	-	-	-	-	-	1	2
3	*feoB/fyuA/fimH*	-	-	-	-	-	-	-	-	-	-	-	-	-	-	-	-	-	2	-	-	-	-	-	-	-	-	-	2
4	*feoB/irp2/fyuA/fimH/traT/* *malX/ompT/iha/kpsMT/usp/* *iroN/hlyA/papA/papGII/* *sfaS/set1/cnf1/focG/iuc*	-	-	-	-	-	-	-	-	-	-	-	1	-	-	-	-	-	-	-	-	-	-	1	-	-	-	-	2
5	*feoB/irp2/fyuA/fimH/traT/* *malX/ompT/iha/kpsMT/usp/hlyA* */papA/papGII/cnf1/iuc*	-	-	1	-	-	-	-	-	-	-	-	1	-	-	-	-	-	-	-	-	-	-	-	-	-	-	-	2
6	*feoB/irp2/fyuA/fimH/traT* */malX/ompT/iha/kpsMT*	-	-	-	-	-	-	-	-	-	-	-	-	-	-	-	-	-	2	-	-	-	-	-	-	-	-	-	2
7–80	Different combination of the same genes	0	0	4	0	0	0	0	0	7	2	5	12	0	0	9	2	0	19	1	2	6	0	0	2	0	0	16	87

**Table 4 microorganisms-11-00045-t004:** Frequency of virulence genes according to biofilm formation in *E. coli*.

Function	Gene	Biofilm (+) (n = 89)			Biofim (−) (n = 11)	*p*-Value	Total (n = 100)
		Weak (n = 27) No. (%)	Moderate (n = 33) No. (%)	Strong (n = 29) No. (%)	Non Producing (n = 11) No. (%)		
Adhesins	*fimH*	17 (63)	23 (70)	20 (69)	8 (73)	0.93	68
	*iha*	10 (37)	12 (36)	11 (38)	5 (45)	0.96	38
	*papA*	4 (15)	5 (15)	9 (31)	0 (0)	0.12	18
	*sfaS*	6 (22)	4 (12)	6 (21)	1 (9)	0.64	17
	*papGII*	3 (11)	4 (12)	6 (21)	0 (0)	0.44	13
	*focG*	2 (7)	3 (9)	4 (13)	0 (0)	0.74	9
	*afa*	2 (7)	2 (6)	0 (0)	1 (9)	0.42	5
	*papGIII*	2 (7)	0 (0)	1 (3)	2 (18)	**0.04**	5
	*papGI*	0 (0)	0 (0)	0 (0)	0 (0)	0	0
	*bmaE*	0 (0)	0 (0)	0 (0)	0 (0)	0	0
Toxins	*usp*	8 (30)	13 (39)	12 (41)	3 (27)	0.73	36
	*hlyA*	3 (11)	5 (15)	11 (37)	0 (0)	**0.01**	19
	*cnf1*	3 (11)	4 (12)	9 (31)	0 (0)	0.06	16
	*set-1*	4 (14)	5 (15)	6 (20)	1 (9)	0.88	16
	*astA*	4 (14)	4 (12)	1 (3)	1 (9)	0.50	10
	*tsh*	0 (0)	1 (3)	1 (3)	1 (9)	0.43	3
Iron Acquisition	*feoB*	23 (85)	27 (81)	27 (93)	9 (81)	0.57	86
	*fyuA*	16 (59)	25 (75)	21 (72)	7 (63)	0.52	69
	*irp2*	18 (67)	22 (67)	19 (66)	7 (63)	1	66
	*iroN*	11 (40)	10 (30)	10 (34)	5 (45)	0.75	36
	*ireA*	3 (11)	12 (36)	4 (13)	2 (18)	0.07	21
	*iuc*	6 (22)	7 (21)	5 (17)	2 (18)	0.96	20
Protectins	*traT*	13 (48)	17 (52)	15 (52)	5 (45)	0.97	50
	*ompT*	9 (33)	11 (33)	14 (48)	4 (36)	0.61	38
	*kpsMT*	10 (37)	10 (30)	11 (37)	4 (36)	0.92	35
	*iss*	4 (15)	4 (12)	0 (0)	0 (0)	0.10	8
PAIs	*malX*	14 (51)	17 (51)	17 (58)	8 (72)	0.65	56

Notes: Significant *p*-values (<0.05) were presented in bold.

**Table 5 microorganisms-11-00045-t005:** Frequency of the serotypes according to phylogroups in the *E. coli* strains.

Serotype	Gingivitis (n = 12) No. of Strains									Moderate Periodontitis (n = 59) No. of Strains									Chronic Periodontitis (n = 29) No. of Strains									*p*-Value	Total (n = 100)
	A	B1	B2	C	D	E	F	CLADE I	Unidentified	A	B1	B2	C	D	E	F	CLADE I	Unidentified	A	B1	B2	C	D	E	F	CLADE I	Unidentified		
	0	0	5	0	0	0	0	0	7	2	5	15	0	0	9	2	0	26	1	2	6	0	1	2	0	0	17		
O25:H4	-	-	1	-	-	-	-	-	-	-	-	3	-	-	-	-	-	3	-	-	3	-	-	-	-	-	4	0.36	14
O12:H-	-	-	1	-	-	-	-	-	3	-	1	-	-	-	-	1	-	5	-	-	-	-	-	-	-	-	1	0.19	12
O?:H-	-	-	-	-	-	-	-	-	-	-	1	-	-	-	1	1	-	2	-	-	1	-	-	-	-	-	2	0.32	8
O2:H39	-	-	-	-	-	-	-	-	-	-	-	3	-	-	-	-	-	1	-	-	1	-	1	-	-	-	-	**0.04**	6
O48:H-	-	-	-	-	-	-	-	-	-	-	-	-	-	-	1	-	-	2	-	-	1	-	-	-	-	-	-	0.14	4
O25:H18	-	-	-	-	-	-	-	-	2	1	-	-	-	-	-	-	-	-	-	-	-	-	-	1	-	-	-	0.20	4
O6:H1	-	-	1	-	-	-	-	-	-	-	-	2	-	-	-	-	-	1	-	-	-	-	-	-	-	-	-	0.42	4
O?:H?	-	-	-	-	-	-	-	-	-	-	-	-	-	-	-	-	-	2	1	-	-	-	-	-	-	-	-	0.30	3
O15:H18	-	-	-	-	-	-	-	-	-	-	-	-	-	-	2	-	-	-	-	-	-	-	-	1	-	-	-	**0.01**	3
O?:H30	-	-	-	-	-	-	-	-	1	-	-	-	-	-	-	-	-	1	-	-	-	-	-	-	-	-	-	1	2
O1:H-	-	-	-	-	-	-	-	-	-	-	-	-	-	-	-	-	-	2	-	-	-	-	-	-	-	-	-	1	2
O1:H7	-	-	-	-	-	-	-	-	-	-	-	-	-	-	-	-	-	1	-	1	-	-	-	-	-	-	-	0.37	2
O16:H5	-	-	-	-	-	-	-	-	-	-	-	1	-	-	-	-	-	-	-	-	-	-	-	-	-	-	1	1	2
O21:H10	-	-	-	-	-	-	-	-	-	-	1	1	-	-	-	-	-	-	-	-	-	-	-	-	-	-	-	0.18	2
O48:H48	-	-	-	-	-	-	-	-	1	1	-	-	-	-	-	-	-	-	-	-	-	-	-	-	-	-	-	0.15	2
O58:H34	-	-	-	-	-	-	-	-	-	-	-	-	-	-	2	-	-	-	-	-	-	-	-	-	-	-	-	0.06	2
O6:H?	-	-	-	-	-	-	-	-	-	-	-	-	-	-	-	-	-	-	-	-	-	-	-	-	-	-	2	1	2
O8:H20	-	-	-	-	-	-	-	-	-	-	1	-	-	-	-	-	-	1	-	-	-	-	-	-	-	-	-	0.37	2
OR:H-	-	-	-	-	-	-	-	-	-	-	-	-	-	-	-	-	-	1	-	-	-	-	-	-	-	-	1	1	2
O?:H21	-	-	-	-	-	-	-	-	-	-	-	1	-	-	-	-	-	-	-	-	-	-	-	-	-	-	-	0.5	1
O?:H5	-	-	-	-	-	-	-	-	-	-	-	-	-	-	-	-	-	-	-	-	-	-	-	-	-	-	1	1	1
O?:H55	-	-	-	-	-	-	-	-	-	-	-	-	-	-	-	-	-	-	-	-	-	-	-	-	-	-	1	1	1
O044037:H?	-	-	-	-	-	-	-	-	-	-	-	-	-	-	-	-	-	-	-	-	-	-	-	-	-	-	1	1	1
O1:H?	-	-	-	-	-	-	-	-	-	-	-	-	-	-	-	-	-	-	-	-	-	-	-	-	-	-	1	1	1
O112ac:H40	-	-	-	-	-	-	-	-	-	-	1	-	-	-	-	-	-	-	-	-	-	-	-	-	-	-	-	0.13	1
O12:H9	-	-	-	-	-	-	-	-	-	-	-	-	-	-	-	-	-	1	-	-	-	-	-	-	-	-	-	1	1
O140:H?	-	-	-	-	-	-	-	-	-	-	-	1	-	-	-	-	-	-	-	-	-	-	-	-	-	-	-	0.5	1
O149:H39	-	-	-	-	-	-	-	-	-	-	-	1	-	-	-	-	-	-	-	-	-	-	-	-	-	-	-	0.5	1
O15:H10	-	-	-	-	-	-	-	-	-	-	-	-	-	-	1	-	-	-	-	-	-	-	-	-	-	-	-	0.24	1
O154:H4	-	-	-	-	-	-	-	-	-	-	-	-	-	-	-	-	-	1	-	-	-	-	-	-	-	-	-	1	1
O185:H?	-	-	-	-	-	-	-	-	-	-	-	-	-	-	1	-	-	-	-	-	-	-	-	-	-	-	-	0.24	1
O185:H16	-	-	-	-	-	-	-	-	-	-	-	-	-	-	-	-	-	-	-	1	-	-	-	-	-	-	-	0.13	1
O20:H?	-	-	-	-	-	-	-	-	-	-	-	-	-	-	-	-	-	-	-	-	-	-	-	-	-	-	1	1	1
O20:H34	-	-	1	-	-	-	-	-	-	-	-	-	-	-	-	-	-	-	-	-	-	-	-	-	-	-	-	0.5	1
O23:H-	-	-	-	-	-	-	-	-	-	-	-	-	-	-	-	-	-	-	-	-	-	-	-	-	-	-	1	1	1
O24:H-	-	-	-	-	-	-	-	-	-	-	-	-	-	-	1	-	-	-	-	-	-	-	-	-	-	-	-	0.24	1
O4:H18	-	-	-	-	-	-	-	-	-	-	-	-	-	-	-	-	-	1	-	-	-	-	-	-	-	-	-	1	1
O6:H31	-	-	-	-	-	-	-	-	-	-	-	1	-	-	-	-	-	-	-	-	-	-	-	-	-	-	-	0.5	1
O62:H30	-	-	-	-	-	-	-	-	-	-	-	-	-	-	-	-	-	1	-	-	-	-	-	-	-	-	-	1	1
OR:H39	-	-	1	-	-	-	-	-	-	-	-	-	-	-	-	-	-	-	-	-	-	-	-	-	-	-	-	0.5	1
OR:H4	-	-	-	-	-	-	-	-	-	-	-	1	-	-	-	-	-	-	-	-	-	-	-	-	-	-	-	0.5	1

O?: could not be determined by serology; OR: rough phenotype, reaction to all antisera; H-**:** did not develop flagellum; H?**:** could not be determined by serology. Notes: Significant *p*-values (<0.05) were presented in bold.

## Data Availability

Not applicable.

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
