# Peer review of "High Virulence and Multidrug Resistance of Escherichia coli Isolated in Periodontal Disease"

_microorganisms, 2022, doi:10.3390/microorganisms11010045_

Round 1

Reviewer 1 Report

The manuscript entitled “High virulence and multidrug resistance of Escherichia coli as emergent colonizer in periodontal diseaseinvestigated the virulome, resistome, phylotype, serotype, and biofilm formation in E. coli strains collected from periodontal pockets of patients experienced periodontitis and gingivitis in a healthcare center in Mexico.

The findings from this study is important for the local dentists and healthcare providers to decide on the appropriate treatment. This study can also provide a precious data source for regional and international studies. The topic of this manuscript is appropriate for this type of journal. The study was well conducted and the manuscript was well written. However, some major and minor points are required to be improved as below:

Major revision comments:

Line 72: please check the references [13-17] and change from [13,14,15,16,17] to [13-17] because the references are in continuous order. There are 5 references relating virulence factors in various anaerobic bacteria cause periodontal disease, please provide more information about the relation of reported virulence factors in anaerobic bacteria and in E. coli.

Line 93-96: The samples were immediately cultured in brain heart infusion broth and subsequently reseeded in the solid culture medium. How many volume (ml) of BHI broth was used? How long does it take between BHI broth culture and solid medium culture? Which method was used for reseeding (streaking or plating)? In addition, how did the authors confirm the collected strains were non-duplicated? Even though the strains were isolated from different patients, there are possibilities that the patients can be infected with the same strains

Line 98-102: Section 2.2: The author mentioned the strains were grown in BHI at 37oC for 12h. Whether the word "strains" means a mixture of different colonies collected from BHI broth or a single colony collected from solid medium culture. Please specify clearly this step as it is important to evaluate the purity of extracted DNA sample.

Line 163-171: Please provide the R code and categorical data matrix including virulence genes, phylogroups, serotypes, etc as described in the supplemental data. According to the journal policy, these data must be publicly available. Please check the Instructions for Authors: https://www.mdpi.com/journal/microorganisms/instructions

Line 385: Please add the discussion about the drawback of this study. The majority of strains were not able to identify the serotypes. Please discuss whether the reason is due to technical limitations or due to the novel serotypes. Please discuss the possibility of applying whole genome sequencing techniques to solve the limitation if it is existed.

Minor revision comments:

Line 71: typos "vaious", please correct

Lines 88-91: section 2.1. For the criteria of periodontitis' diagnosis, did the author include the dental X-rays' image. Please include the criteria definitions for moderate periodontitis and chronic periodontitis

Line 94: please add city where the Bioxon is located

Line 96: please add city where the MCD LAB is located

Line 107: 3 uL DNA template (100 ng). What is the stock DNA concentration? Is it 33.333333ng/uL.? how did the author make sure that 100ng were added and equal among samples?

Line 118: please add the city where the company "Investigación Diagnóstica is located

Line 173-183: How did the authors identify the other bacteria associated with E. coli. In the method part, the authors did not mention the methods related to other species identification.

Line 256: Please put table 5 in another page with landscape orientation.

Line 273: Figure 1: Please provide a higher quality figure with dpi at 600 dpi. Please check the Instructions for Authors: https://www.mdpi.com/journal/microorganisms/instructions

Line 277: Please add "," after "rectangle"

Line 279: Please add "," after "left"

Line 317: Please do not italicize "genes"

Author Response

Manuscript

microorganisms- 2098473

High virulence and multidrug resistance of Escherichia coli isolated in

periodontal disease.

Microorganisms

Response letter

We are thankful to the reviewers for this opportunity to improve our manuscript. We appreciate the insightful and constructive comments, and the time spent on their review. In this revised version we addressed all review comments and have modified the text accordingly. These additions, clarifications and modifications have improved significantly the quality of the manuscript. The following reply addresses these important issues point by point. Our answers are written in bold script.

Reviewer comments 1:

Major revision comments:

Line 72: please check the references [13-17] and change from [13,14,15,16,17] to [13-17] because the references are in continuous order. There are 5 references relating virulence factors in various anaerobic bacteria cause periodontal disease, please provide more information about the relation of reported virulence factors in anaerobic bacteria and in E. coli.

Thank you for the observation and for this valuable comment. We corrected the references, and added a new paragraph in the Introduction section that describes the virulence factors of the anaerobic bacteria that cause periodontal disease, as well as the virulence factors of E. coli.

Line 93-96: The samples were immediately cultured in brain heart infusion broth and subsequently reseeded in the solid culture medium. How many volume (ml) of BHI broth was used? How long does it take between BHI broth culture and solid medium culture? Which method was used for reseeding (streaking or plating)? In addition, how did the authors confirm the collected strains were non-duplicated? Even though the strains were isolated from different patients, there are possibilities that the patients can be infected with the same strains

Thank you for the comment. In the Materials and methods section, we have clarified and expanded in detail the methods of culturing the samples on the solid media, as well as the cross-streaking method that was used in bacterial reseeding.

Using the unsupervised hierarchical clustering analysis (figure 1), we confirmed that 80% of the strains obtained from each patient were

different, because they presented a distinct array of virulence and genotype. However, in agreement with your important observation, we detected 20% of strains with the same virulence and genotype composition. We included a descriptive paragraph in the Results section and we elaborate on this in the Discussion section.

Line 98-102: Section 2.2: The author mentioned the strains were grown in BHI at 37oC for 12h. Whether the word "strains" means a mixture of different colonies collected from BHI broth or a single colony collected from solid medium culture. Please specify clearly this step as it is important to evaluate the purity of extracted DNA sample.

Thank you for this valuable comment. In the Material and methods section, we clarify and explain in detail that from a pure culture of each E. coli strain identified by standard IMViC biochemical tests, a colony was taken, reseeded in 2 mL of Brain Heart Infusion (BHI) broth and incubated at 37 °C for 12 h.

Line 163-171: Please provide the R code and categorical data matrix including virulence genes, phylogroups, serotypes, etc as described in the supplemental data. According to the journal policy, these data must be publicly available. Please check the Instructions for Authors: https://www.mdpi.com/journal/microorganisms/instructions

Thank you for your kind observation. We provide the R codes and data matrices used as supplementary material. We referenced these materials in the materials and methods section.

Line 385: Please add the discussion about the drawback of this study. The majority of strains were not able to identify the serotypes. Please discuss whether the reason is due to technical limitations or due to the novel serotypes. Please discuss the possibility of applying whole genome sequencing techniques to solve the limitation if it is existed.

Thank you for the observation and for this valuable comment. In the Discussion section we added a paragraph where we provide possible explanations of why the phylogroups were not identified in half of the strains, and also why some serotypes could not be identified in the strains. We added two references to support this discussion. We agree that whole genome sequencing would surpass this limitation. This sentence was also added to the Discussion section.

Minor revision comments:

Line 71: typos "vaious", please correct

Thanks, we made this correction.

Lines 88-91: section 2.1. For the criteria of periodontitis' diagnosis, did the author include the dental X-rays' image. Please include the criteria definitions for moderate periodontitis and chronic periodontitis

Thanks for the observation. Radiographs were taken to the patients to establish the criteria for periodontitis. We included the criteria for moderate and chronic periodontitis in the Material and methods section.

Line 94: please add city where the Bioxon is located

Thank you. We added the city.

Line 96: please add city where the MCD LAB is located

Thank you. We added the city.

Line 107: 3 uL DNA template (100 ng). What is the stock DNA concentration? Is it 33.333333ng/uL.? how did the author make sure that 100ng were added and equal among samples?

Thank you for your kind observation. In the material and methods section, we have clarified in detail that Stock dilutions of genomic DNA from each E. coli strain were prepared with nuclease-free water to obtain concentrations of 100 ng/μL. We also clarify that the final concentration of DNA used for each PCR reaction was 300 ng.

Line 118: please add the city where the company "Investigación Diagnóstica” is located

Thank you. We added the city.

Line 173-183: How did the authors identify the other bacteria associated with E. coli. In the method part, the authors did not mention the methods related to other species identification.

Thank you for your valuable comment. We included in the Materials and methods section the molecular methods for the identification of other microorganisms associated with E. coli.

Line 256: Please put table 5 in another page with landscape orientation.

Thank you for the observation. We changed the orientation of the table as you suggest.

Line 273: Figure 1: Please provide a higher quality figure with dpi at 600 dpi. Please check the Instructions for Authors: https://www.mdpi.com/journal/microorganisms/instructions

Thanks for the observation. We will upload a figure with better quality and resolution separately from the manuscript.

Line 277: Please add "," after "rectangle"

Thank you. We made this correction.

Line 279: Please add "," after "left"

Thank you. We made this correction.

Line 317: Please do not italicize "genes"

Thank you. We made this correction.

Reviewer 2 Report

The authors provided a good insight into the issue of other potential pathogens in periodontal diseases, E. coli in this case. Though E. coli seems to be a secondary pathogen inducing the oral diseases, its infectivity and severity should not be ignored. This study also highlighted the antibiotic-resistant problem. Overall, the study is brilliant, and the authors tries to present it well and carefully. However, English editing might be needed to make it better. Also, some descriptions would need to be improve. Please see the following comments:

-Introduction

1. It could be better to rearrange/reorganize the section. Paragraphs are like bits and pieces.

2. Not enough introduction. Please add a paragraph to introduce the current situation, for example PMID: 26416306.

-Material and methods

3. biofilm formation assay. The interpretation criteria seem to be based on the following formula according to the reference you cited. Please clarify.

“Biofilm formation capacity is equal to stained attached bacterial divided by negative control.” (BF=AB/CW)

4. Please explain the statistical analysis for all results.

-Results

5. Line 178-181. Since the usage of the EMB plates for isolation of E. coli is claimed in Methods section, the isolation of Gram-positive bacteria makes no sense.

6. Line 185-186. Please use full name for antibiotics in main text to help readers better understand.

7. Table 1. Please add a head “No. of resistant isolates (%)” above the three groups.

8. Line 213- and table 3. The part is not well interpreted (hard to understand). For example, eight patterns are claimed but the Table 3 shows 1, 2, 3, 4, 5, 6, and 7-80 (don’t even know what 7-80 is); “(n=19)”, which is stand for 19 virulence genes, is noticed in main but the presentation way like this easily mislead readers. Please improve the descriptions.

9. Table 4. Why is the number in weak producer and fimH gene (17 (63)) in bold?

10. Line 252-253. All isolates with serotype O15:H18 are belonging to phylogroup E not D. Please check the results or revise the description.

11. Figure 1. Well done!

-Discussion

12. Line 293-. Same as point 5. Please use full name for antibiotics in main text.

-Shortcomings

13. Big pities not to see any results about ESBL detection (genetic or phenotypic), pulsed filed gel electrophoresis, and multi-locus sequence type.

Author Response

Manuscript microorganisms- 2098473

High virulence and multidrug resistance of Escherichia coli isolated in periodontal disease.

Microorganisms

Response letter

We are thankful to the reviewers for this opportunity to improve our manuscript. We appreciate the insightful and constructive comments, and the time spent on their review. In this revised version we addressed all review  comments and have modified the text accordingly. These additions, clarifications and modifications have improved significantly the quality of the manuscript. The following reply addresses these important issues point by point. Our answers are written in bold script.

Reviewer comments 2: 

Comments and Suggestions for Authors

The authors provided a good insight into the issue of other potential pathogens in periodontal diseases, E. coli in this case. Though E. coli seems to be a secondary pathogen inducing the oral diseases, its infectivity and severity should not be ignored. This study also highlighted the antibiotic-resistant problem. Overall, the study is brilliant, and the authors tries to present it well and carefully. However, English editing might be needed to make it better. Also, some descriptions would need to be improve. Please see the following comments:

-Introduction

  1. It could be better to rearrange/reorganize the section. Paragraphs are like bits and pieces.

Thank you for the observation and for your valuable comment. The Introduction was improved.

  1. Not enough introduction. Please add a paragraph to introduce the current situation, for example PMID: 26416306.

Thank you for your kind suggestion. We have included a paragraph with the information from the suggested article "PMID: 26416306". The introduction was also expanded with more information on the virulence factors of the anaerobic bacteria that cause periodontal disease.

-Material and methods

  1. biofilm formation assay. The interpretation criteria seem to be based on the following formula according to the reference you cited. Please clarify.

“Biofilm formation capacity is equal to stained attached bacterial divided by negative control.” (BF=AB/CW)

Thank you for the observation. In the Materials and methods section, we have clarified in detail the interpretation criteria for biofilm formation in E. coli strains.

  1. Please explain the statistical analysis for all results.

Thank you for the observation. In the material and methods section we include a paragraph to explain in detail the statistical analysis performed on all the results.

-Results

  1. Line 178-181. Since the usage of the EMB plates for isolation of E. coliis claimed in Methods section, the isolation of Gram-positive bacteria makes no sense.

Thank you very much for your valuable comment. Since other reviewers have suggested that to improve understanding of E. coli associated with S. aureus, C. albicans, and K. pneumoniae during periodontal disease, culture media for identifying these microorganisms are included in materials and methods.

  1. Line 185-186. Please use full name for antibiotics in main text to help readers better understand.

Thanks for the observation. We included the full name of all antibiotics.

  1. Table 1. Please add a head “No. of resistant isolates (%)” above the three groups.

Thank you. We have made the correction in table 1.

  1. Line 213- and table 3. The part is not well interpreted (hard to understand). For example, eight patterns are claimed but the Table 3 shows 1, 2, 3, 4, 5, 6, and 7-80 (don’t even know what 7-80 is); “(n=19)”, which is stand for 19 virulence genes, is noticed in main but the presentation way like this easily mislead readers. Please improve the descriptions.

Thank you for the observation and for your valuable comment. In the Results section, in table 3, we improved, expanded and describe in detail the different association combinations of virulence genes to facilitate the reading of the information.

We also clarified that there were 87 strains in the virulence gene combinations of No. 7 to 80. Each of these strains presented a different combination of genes.

  1. Table 4. Why is the number in weak producer and fimHgene (17 (63)) in bold?

Thank you for the observation. We are sorry for the confusion. We think that an error occurred during the transformation of our manuscript to PDF format or was an unintentional error of ours. The bold letters have been changed.

  1. Line 252-253. All isolates with serotype O15:H18 are belonging to phylogroup E not D. Please check the results or revise the description.

Thank you for your valuable observation. We are sorry for the mistake. We made this correction.

  1. Figure 1. Well done!

Thank you so much for the compliment! We appreciate it!

-Discussion

  1. Line 293-. Same as point 5. Please use full name for antibiotics in main text.

Thank you for the observation. We included the full name of all antibiotics.

-Shortcomings

  1. Big pities not to see any results about ESBL detection (genetic or phenotypic), pulsed filed gel electrophoresis, and multi-locus sequence type.

Thanks for your kind comment. Currently, the analysis of the strains by electrophoresis in pulsed fields is being carried out, it also began with the identification of antibiotic resistance genes and soon we will begin with the expression of virulence markers using oral epithelial cell lines. Hopefully, soon we will have enough information to send a next article to this prestigious journal.

Reviewer 3 Report

General comment: This work aimed to determine the variability in the virulence genotype, antibiotic resistance phenotype, biofilm formation, phylogroups, and serotypes in the strains of E. coli isolated from patients with periodontal disease. This is an interesting work focusing in the characterization of a bacterial species rarely identified in periodontal disease.

1.       Missing references in lines 48, 110, 139, 149, 159, 191.

Specific comments:

2.       The abstract:  the suggestion of E. coli as a secondary colonizer is not evaluated in this work. You only characterized the E. coli isolates, you do not study the ability of E.coli to co-aggregate with early colonizers of the enamel. Please modify this sentence (line 39) and all parts when you mention that E. coli is a secondary colonizer.

3.       Introduction: Attention that in Periodontal disease also participate Gram positive bacteria, please include that in the introduction section.

4.       The material and methods section:

4.1   Please include the primers concentration, it is 10 pmol? It is not clear to the readers ( lines 103-107; 126-130;144-146).

4.2   Please include as annex or supplementary file information regarding all genes submitted to PCR, including sequences, concentrations, references used. Considering Virulence genes and phylogroups.

4.3   A statistical analysis heading should be included, with the information present in lines 156-159.

5.       Results Section:

5.1   How did you perform the multidrug resistance classification of the isolates? This must be included in the methods section with the correct reference.

5.2   Include the abbreviation of the antibiotics on the table 1.

Author Response

Manuscript

microorganisms- 2098473

High virulence and multidrug resistance of Escherichia coli isolated in

periodontal disease.

Microorganisms

Response letter

We are thankful to the reviewers for this opportunity to improve our manuscript. We appreciate the insightful and constructive comments, and the time spent on their review. In this revised version we addressed all review comments and have modified the text accordingly. These additions, clarifications and modifications have improved significantly the quality of the manuscript. The following reply addresses these important issues point by point. Our answers are written in bold script.

Reviewer comments 3:

Comments and Suggestions for Authors

General comment: This work aimed to determine the variability in the virulence genotype, antibiotic resistance phenotype, biofilm formation, phylogroups, and serotypes in the strains of E. coli isolated from patients with periodontal disease. This is an interesting work focusing in the characterization of a bacterial species rarely identified in periodontal disease.

Missing references in lines 48, 110, 139, 149, 159, 191.

Thank you for your kind observation. We have included the missing references in the manuscript.

Specific comments:

The abstract: the suggestion of E. coli as a secondary colonizer is not evaluated in this work. You only characterized the E. coli isolates, you do not study the ability of E.coli to co- aggregate with early colonizers of the enamel. Please modify this sentence (line 39) and all parts when you mention that E. coli is a secondary colonizer.

Thank you for the observation and for your valuable comment. We have modified the sentences throughout the text where E. coli is mentioned as a secondary colonizer, including the title of the manuscript.

Introduction: Attention that in Periodontal disease also participate Gram positive bacteria, please include that in the introduction section.

We thank you very much for your valuable comment. In the Introduction section, we included the Gram-positive bacteria that also participate in periodontal disease.

4. The material and methods section:

4.1 Please include the primers concentration, it is 10 pmol? It is not clear to the readers ( lines 103-107; 126-130;144-146).

Thank you for the observation. We include the concentration of the primers (10 pmol) in the Materials and methods section.

4.2 Please include as annex or supplementary file information regarding all genes submitted to PCR, including sequences, concentrations, references used. Considering Virulence genes and phylogroups.

Thank you for the observation. We include as a supplementary file the information of all the genes evaluated by PCR, the sequences of the primers, the size of the amplicons, and the references.

4.3 A statistical analysis heading should be included, with the information present in lines 156- 159.

We appreciate the kind comment. We have included a statistical analysis header.

5. Results Section:

5.1 How did you perform the multidrug resistance classification of the isolates? This must be included in the methods section with the correct reference.

Thank you for the observation. We have clarified the criteria for classifying multidrug resistant strains in the Material and methods section. A reference has been included.

5.2 Include the abbreviation of the antibiotics on the table 1.

Thank you for this comment. We have included the abbreviation of the antibiotics in Table 1.

Round 2

Reviewer 1 Report

Thank you for your responses to my comments. There is a minor comment

Line 208: the word “serotype” was duplicated. Please check